# Cell-DETR: Efficient cell detection and classification in WSIs with transformers

**Oscar Pina**[1]                   OSCAR.PINA@UPC.EDU
[1] *Universitat Politècnica de Catalunya - BarcelonaTech (UPC)*

**Eduard Dorca**[2]              EDORCA@HOSPITALBELLVITGE.CAT
[2] *Hospital Universitari de Bellvitge (HUB)*

**Verónica Vilaplana**[1]            VERONICA.VILAPLANA@UPC.EDU

**Editors:** Accepted for publication at MIDL 2024

## Abstract

Understanding cell interactions and subpopulation distribution is crucial for pathologists to support their diagnoses. This cell information is traditionally extracted from segmentation methods, which poses significant computational challenges on processing Whole Slide Images (WSIs) due to their giga-size nature. Nonetheless, the clinically relevant tasks are nuclei detection and classification rather than segmentation. In this manuscript, we undertake a comprehensive exploration of the applicability of detection transformers for cell detection and classification (Cell-DETR). Not only do we demonstrate the effectiveness of the method by achieving state-of-the-art performance on well-established benchmarks, but we also develop a pipeline to tackle these tasks on WSIs at scale to enable the development of downstream applications. We show its efficiency and feasibility by reporting a $\times 3.4$ faster inference time on a dataset featuring large WSIs. By addressing the challenges associated with large-scale cell detection, our work contributes valuable insights that paves the way for the development of scalable diagnosis pipelines based on cell-level information.

**Keywords:** Object detection, cell detection, transformers

## 1. Introduction

The integration of deep learning methods into digital pathology image analysis is reshaping medical practices, offering unprecedented opportunities for enhanced diagnostics. These techniques span from analyzing individual cells to examining Whole Slide Images (WSIs). Pathologists often rely on the composition of diverse cell subtypes, and other biological entities such as glands, in order to support their diagnoses, making the precise identification of cell nuclei imperative for effective computer-aided diagnosis applications. Indeed, applications leveraging cell information are gaining considerable attention (Pati et al., 2022; Jaume et al., 2021).

Cell segmentation and classification represent well-explored tasks in digital pathology (Graham et al., 2019; Hörst et al., 2023), supported by various datasets for related research (Gamper et al., 2020; Graham et al., 2019). However, the truly clinically relevant objectives lie in cell instance detection and classification, prioritizing these over segmentation. The inclination towards segmentation as a surrogate for detection arises from the inherent challenges posed by the size, morphology, and density of cell nuclei. Given their small size and frequent overlap, direct detection becomes a complex task. Additionally, accurate

classification often relies on subtle image details, necessitating high resolutions for robust outcomes. Segmentation methods excel in capturing those small details, contributing to improved results.

Despite the valuable boost in accuracy, this improvement comes at a cost—significant computational demands during both training and inference. The dense output format of segmentation masks amplifies memory and computational resource requirements for calculating pixel-level predictions and training loss function. Moreover, it also involves expensive post-processing steps during inference to output the final predictions. The size of WSIs, often reaching gigapixel dimensions (e.g. $100,000 \times 100,000 = 10^{10}$ pixels), adds an extra layer of complexity to implementing a computer-aided diagnosis pipeline, requiring the partition into smaller patches and subsequent processing. This challenge is particularly pronounced for segmentation methods, given their dense pixel-level output maps, making them impractical for real-world applications on WSIs. Instead, their application is often limited to smaller tiles, hindering the development of computer-aided diagnoses that require comprehensive cell information.

In this study, we delve into the challenges of cell nuclei detection and classification, treating it as a traditional object detection task. We explore the opportunities and challenges associated with this approach, presenting novel insights and methodologies geared towards enhancing the efficiency and accuracy of this critical facet of digital pathology image analysis and extending their application to WSIs.

To develop large-scale methods for detecting and classifying cell nuclei, we utilize the DEtection TRansformer (DETR) model (Carion et al., 2020; Zhu et al., 2020). Although earlier studies have explored DETRs for cell detection and classification in digital pathology (Obeid et al., 2022; Huang et al., 2023), we prioritize practicality and robustness of cell detection transformers (Cell-DETR) rather than focusing on designing sophisticated auxiliary architectures as in prior methodologies. Our key objective is to provide the necessary tools to integrate Cell-DETR into daily clinical workflows by firstly obtaining more reliable, robust models and secondly addressing the challenges that arise when applying them to real world, large-scale scenarios beyond the traditional patch-based datasets.

Our contributions are twofold: firstly, to enhance the reliability of Cell-DETR, we explore different design components of DETR models and achieve state-of-the-art performance in both cell detection and classification tasks on popular benchmarks. Subsequently, we derive a specialized pipeline for efficient inference on WSIs, achieving a remarkable $\times 3.4$ speed-up on inference time compared to other methods. This enhancement significantly expedites the application of Cell-DETR models to WSIs, making them well-suited for large-scale digital pathology tasks with cell-level information.

This manuscript is structured as follows: In Section 2, we provide the essential background information and related work to our topic. Section 3 outlines our methodologies, including details on datasets, augmentations, architecture and the inference pipeline designed for WSIs. The evaluation of our models for cell detection and classification, alongside the measurement of inference time on WSIs, is presented in Section 4. Finally, our conclusions are summarized in Section 5.

## 2. Background and related work

### 2.1. Cell segmentation and classification

HoVer-Net (Graham et al., 2019) introduces an innovative U-Net-like architecture featuring three decoder branches: nuclear pixel (NP), horizontal-vertical (HV), and nuclear classification (NC). These branches play distinct roles in predicting the probability of a pixel belonging to a nucleus, the horizontal and vertical distances to the nucleus's center of mass, and the pixel label, respectively. A postprocessing step is required to merge the outputs of the NP and HV branches to generate the final segmentation mask.

A recent extension of this work, CellViT (Hörst et al., 2023), takes a step further by replacing the convolutional encoder with a Vision Transformer (ViT) (Dosovitskiy et al., 2021), achieving state-of-the-art performance in cell detection and classification. This transition to a transformer-based architecture showcases the adaptability and effectiveness of transformer models in the domain of medical image analysis (You et al., 2022).

### 2.2. Object detection with Transformers

The Detection Transformer (DETR) (Carion et al., 2020) presents an end-to-end approach to object detection, utilizing transformers and bipartite matching to eliminate the necessity for manual post-processing steps. The model consists of a backbone that extracts hidden features from an input image, a transformer encoder that enhances these features through self-attention, and a transformer decoder that given the encoded image information outputs bounding box predictions for a set of input queries, which are learnable parameters of the model. The model undergoes training with a set-based bipartite matching loss to ensure the uniqueness of predictions.

DETR exhibits certain limitations, particularly in its ability to detect small objects due to the global nature of self-attention. To address this constraints, Deformable-DETR (Zhu et al., 2020) incorporates a multi-scale deformable attention operation that confines the attention of each token to a specific subset of points. The determination of these points is achieved through the prediction of multi-scale offsets from the central token position to other tokens across all scales. Importantly, this offset prediction is co-trained with other components of the model, providing a comprehensive approach to enhancing the model's performance in detecting smaller objects. This approach also brings additional advantages, including multi-scale representations and faster computation.

### 2.3. Cell detection and classification with transformers

The exploration of cell detection and classification using transformers has been a subject of previous research, as evident in related works such as NucDETR (Obeid et al., 2022) and ACFormer (Huang et al., 2023). NucDETR (Obeid et al., 2022) stands out as the pioneering work that introduced the application of DETR for cell detection. However, it did not delve into the task of nuclei classification.

On the other hand, ACFormer (Huang et al., 2023) presents a sophisticated mechanism featuring an adaptive transformer. This transformer proposes affine transformations for a given input image to be used as data augmentation. The method incorporates a local-global network architecture and a self-distillation mechanism, where the local network receives the

affine-transformed images as input whereas the global network is fed with the original image. The outputs of the global network serve as target for the local network. While intriguing, the proposed transformations and the advantages of its local-global strategy remain unclear, suggesting at an excessively complex approach that may compromise the practicality and reliability of the model.

## 3. Methods

### 3.1. Datasets

**PanNuke**    The PanNuke dataset (Gamper et al., 2020) comprises 7,904 patches, each sized $256 \times 256$, extracted from WSIs in The Cancer Genome Atlas (TCGA) dataset, representing 19 diverse tissue types at a magnification of 40x. Within this dataset, there are 189,744 labeled nuclei categorized into five clinically significant classes: neoplastic, inflammatory, connective, necrosis, and epithelial.

**CoNSeP**    The CoNSeP dataset (Graham et al., 2019) includes 41 $1000 \times 1000$px tiles extracted from H&E-stained colorectal adenocarcinoma WSIs, each with at a 40x magnification. Notably diverse, the dataset encompasses various regions such as stromal, glandular, muscular, collagen, adipose, and tumorous areas. It also features a range of nuclei derived from different cell types, which are grouped into inflammatory, epithelial, spindle-shaped and miscellaneous (Graham et al., 2019).

**Camelyon16**    The Camelyon16 dataset (Bejnordi et al., 2017) consists of 400 H&E stained Whole Slide Images (WSIs) of lymph node sections scanned at $\times 40$ magnification. Each WSI is accompanied by annotations highlighting tumor and normal regions. With average dimensions of $189,832 \times 95,590$px, approximately 29% of the slides represent tissue area. Specifically, an average of 1384 tissue tiles, each sized $2048 \times 2048$ pixels, is extracted per slide. Despite lacking cell-level annotations for quantifying detection and classification performance, this dataset remains pivotal in evaluating the efficacy of our models. Leveraging the Camelyon16 dataset allows us to assess the practicality of Cell-DETR on a scale that closely mimics the challenges encountered in clinical settings, demonstrating their scalability and effectiveness in handling large-scale pathology images.

### 3.2. Cell-DETR

**Architecture**    The architecture for Cell-DETR comprises a hierarchical backbone that generates a four level feature pyramid for a given input image, followed by a multi-scale deformable transformer (Zhu et al., 2020), consisting of 6 encoder and 6 decoder layers. The encoder enhances the input features through multi-scale deformable self-attention, while the decoder produces predictions for bounding boxes and labels based on a set of input object queries. The initial states of these queries are foreground proposals generated from the output of the encoder (Zhu et al., 2020). Both the backbone and the transformer are pretrained on the COCO dataset (Lin et al., 2014). In Section 4.1, we conduct experiments using both ResNet-50 (He et al., 2016) and Swin Transformer (Liu et al., 2021) backbones. Additionally, in the Appendix C.1 we explore the impact of the output resolution and the number of levels in the extracted image features.

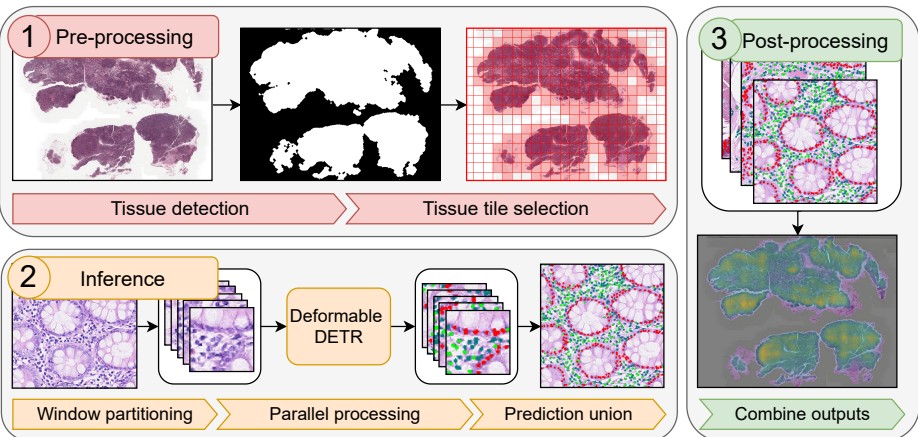

Figure 1: Cell-DETR pipeline for efficient cell detection on WSIs.
*(1) Preprocessing:* The tissue area of the slide is identified, and the slide is segmented into tiles of size 2048 × 2048px. *(2) Inference:* Each tile undergoes the *window detection* procedure.The tiles are fed into the model, divided into overlapping windows, processed in parallel, and their predictions are merged. *(3) Post-processing:* The outputs from all tiles are aggregated to derive the cell nuclei for the WSI. The figure depicts a heatmap illustrating the cell density.

**Data augmentation**  Data augmentation plays a crucial role in the domain of digital pathology. Images exhibit substantial diversity due to various factors, including differences in staining protocols, elapsed time since slide staining before digitization, and the diverse tissue types. Acknowledging and addressing these variations through data augmentation is key for obtaining robust performance across different conditions. Drawing from the observations in (Tellez et al., 2018), our data augmentation pipeline includes not only traditional rotation, flipping, and blurring augmentations but also a combination of elastic transformation and stain augmentations. The latter involves transforming the RGB image into the Hematoxylin-Eosin-DAB space (HED), randomly corrupting the channels separately, and then transforming the image back to the RGB space.

**Loss function**  We utilize the standard loss function recommended for DETRs in natural images, which includes a combination of bounding box L1 regression, generalized intersection over union and focal loss classification. Opposed to ACFormer (Huang et al., 2023), which limits its prediction to the nuclei centroids as it is the primary interest in cell detection, we have observed a slight performance decline when excluding the boxes' width and height from the loss computation. Our hypothesis is that incorporating feedback on the boxes size during training aids the network in disambiguating detections and predicting the class label. The focal loss is used for classification rather than the standard cross-entropy loss to account for the class imbalance between cell nuclei. The corresponding hyperparameters and the difference in performance when excluding the bounding box size from the target can be found in the Appendinx A and the Appendix C.2, respectively.

### 3.3. Large-scale cell detection

Large-scale cell detection and classification on WSIs poses a formidable challenge owing to their gigapixel size. Traditional segmentation approaches might face significant hurdles in

terms of computational and memory resources, making them less practical for addressing this task. In this section, we explore an alternative approach by Cell-DETR for processing on larger tiles. By adopting a strategic *window detection* procedure, we facilitate the application of Cell-DETR on larger images, enabling efficient and scalable inference on WSIs. This approach provides a practical solution for large-scale cell detection and classification tasks overcoming the massive scale of histopathological images.

### 3.3.1. Dealing with large image tiles

A significant constraint of DETR-like models is the necessity for the number of queries in the decoder to surpass the potential objects present in an image. In regions characterized by a high cell density, a $256 \times 256$px image patch may contain up to 300 cell nuclei. Consequently, increasing the input image size to larger tiles, such as $1024 \times 1024$px or $2048 \times 2048$px, becomes non-trivial. The number of cell nuclei, and therefore the required input DETR queries, can substantially increase, potentially resulting in prohibitive computational demands.

To address this challenge when working with larger image tiles, we adopt a *window detection* procedure. This involves training the model using randomly selected image crops of the desired size extracted from the original images. During inference, an overlapped sliding window approach is adopted, which is executed in-device to minimize GPU-CPU communication and enhance inference speed. Concretely, the model splits the original image into overlapped windows, processes them in parallel and finally combines the outputs to derive the final results. This strategy allows Cell-DETR to overcome the limitations associated with regions containing a high density of cell nuclei and large images, ensuring the model's adaptability and efficiency in real-world applications.

### 3.3.2. Inference on WSIs

Scalability of cell detection and classification on WSIs is central to our approach, driven by the giga-size of these images. We have devised a robust pipeline tailored for inference on WSIs leveraging the *window detection* approach. Specifically, the tissue regions of the slide are subdivided into $2048 \times 2048$ tiles, and these tiles undergo processing using the *window detection* procedure. Subsequently, all predictions are aggregated to obtain the final results. This approach is more suitable than directly partitioning the image into smaller patches that could directly be fed into the model. The in-device execution of the *window detection* ensures an efficient and streamlined process, making our pipeline adept at handling the distinctive challenges associated with the substantial scale of WSIs.

Table 1: Detection and classification metrics on PanNuke dataset.

| Method | Detection | | | Neoplastic | | | Epithelial | | | Inflammatory | | | Connective | | | Necrosis | | |
|---|---|---|---|---|---|---|---|---|---|---|---|---|---|---|---|---|---|---|
| | $P_{det}$ | $R_{det}$ | $F_{det}$ | $P_{neo}$ | $R_{neo}$ | $F_{neo}$ | $P_{epi}$ | $R_{epi}$ | $F_{epi}$ | $P_{inf}$ | $R_{inf}$ | $F_{inf}$ | $P_{con}$ | $R_{con}$ | $F_{con}$ | $P_{nec}$ | $R_{nec}$ | $F_{nec}$ |
| **DIST** (Naylor et al., 2018) | 0.74 | 0.71 | 0.73 | 0.49 | 0.55 | 0.50 | 0.38 | 0.33 | 0.35 | 0.42 | 0.45 | 0.42 | 0.42 | 0.37 | 0.39 | 0.00 | 0.00 | 0.00 |
| **Mask-RCNN** (He et al., 2017) | 0.76 | 0.68 | 0.72 | 0.55 | 0.63 | 0.59 | 0.52 | 0.52 | 0.52 | 0.46 | 0.54 | 0.50 | 0.42 | 0.43 | 0.42 | 0.17 | 0.30 | 0.22 |
| **Micro-Net** (Raza et al., 2019) | 0.78 | 0.82 | 0.80 | 0.59 | 0.66 | 0.62 | 0.63 | 0.54 | 0.58 | 0.59 | 0.46 | 0.52 | 0.40 | 0.45 | 0.47 | 0.23 | 0.17 | 0.19 |
| **HoVerNet** (Graham et al., 2019) | 0.82 | 0.79 | 0.80 | 0.58 | 0.67 | 0.62 | 0.54 | 0.60 | 0.56 | 0.56 | 0.51 | 0.54 | 0.52 | 0.47 | 0.49 | 0.28 | 0.35 | 0.31 |
| **CellViT** (Hörst et al., 2023) | 0.83 | 0.82 | **0.82** | 0.69 | 0.70 | 0.69 | 0.68 | 0.71 | 0.70 | 0.59 | 0.58 | 0.58 | 0.53 | 0.51 | 0.52 | 0.39 | 0.35 | 0.37 |
| **Cell-DETR R50** | 0.85 | 0.78 | 0.81 | 0.72 | 0.67 | 0.69 | 0.71 | 0.67 | 0.69 | 0.59 | 0.60 | 0.59 | 0.57 | 0.49 | 0.53 | 0.54 | 0.32 | 0.40 |
| **Cell-DETR SwinL** | 0.85 | 0.80 | **0.82** | 0.74 | 0.70 | **0.72** | 0.74 | 0.74 | **0.74** | 0.60 | 0.63 | **0.61** | 0.60 | 0.52 | **0.56** | 0.56 | 0.41 | **0.47** |

Other metrics are extracted from (Hörst et al., 2023).

Table 2: Detection and classification F-Score on CoNSeP dataset.

| Method | Detection | Epithelial | Inflammatory | Spindle-shaped | Miscellaneous |
|---|---|---|---|---|---|
| **DIST** (Naylor et al., 2018) | 0.71 | 0.62 | 0.53 | 0.51 | 0.00 |
| **Micro-Net** (Raza et al., 2019) | 0.74 | 0.62 | 0.59 | 0.53 | 0.12 |
| **Mask-RCNN** (He et al., 2017) | 0.69 | 0.60 | 0.59 | 0.52 | 0.10 |
| **HoVer-Net** (Graham et al., 2019) | **0.75** | 0.64 | 0.63 | **0.57** | **0.43** |
| **ACFormer** (Huang et al., 2023) | 0.74 | 0.64 | 0.64 | - | - |
| **Cell-DETR R50** | 0.74 | 0.61 | 0.63 | 0.51 | 0.21 |
| **Cell-DETR SwinL** | 0.74 | **0.65** | **0.67** | 0.56 | 0.40 |
| **Cell-DETR SwinL*** | *0.77* | *0.70* | *0.70* | *0.61* | *0.55* |

*Model pre-trained on the first fold of PanNuKe dataset.

## 4. Results

In this section, we conduct an extensive set of experiments and present the results to provide insights into the capabilities of Cell-DETR. We assess the performance in terms of F-Score and inference time in Section 4.1 and Section 4.2, respectively.

### 4.1. Detection and classification performance

Table 1 presents the detection and classification metrics on the PanNuke dataset for Cell-DETR using ResNet-50 and Swin Transformer (large) as backbones, in comparison to other state-of-the-art segmentation and detection methods. The provided numerical values in the table detail the averaged precision (P), recall (R), and F1-score (F1) for detection and nuclei types across the three standard splits publicly available for this dataset. A more detailed description of the metrics can be found in Appendix B. Notably, our results align with the current state-of-the-art in cell detection, and we achieve state-of-the-art performance in cell classification. The ResNet-50 backbone exhibits slightly superior classification performance, while the Swin-L backbone surpasses classification metrics by a significant margin. Although including the Swin-L involves an increase in the parameter complexity, these results showcase the potential of transformers for medical image analysis.

Given that the CoNSeP dataset consists of tiles sized at $1000 \times 1000$ pixels, the potential number of nuclei in a single image is exceptionally high. We have employed the *window detection* procedure outlined in Section 3.3 for both training and evaluating the Cell-DETR on this dataset. For training, random crops of $250 \times 250$px are randomly samples. During the valuation phase, the tiles are processed with the *window detection*, with a window size of 250px and a stride of 187. The resulting predictions are then combined, retaining only those detections within the central crop of 187x187 pixels for each window. The detection and classification F-Score results are presented in Table 2, showcasing state-of-the-art performance. These findings align with the results presented in Table 1 and validate the effectiveness of the *window detection* approach in handling scenarios with high nucleus abundance.

### 4.2. Time performance

To evaluate the scalability and feasibility of Cell-DETR models, along with the *window detection* procedure, we conduct experiments on a subset of 111 slides from the Camelyon16 dataset and compare the time performance with HoVer-Net. The inference process can be divided into three steps: (i) model loading and slide pre-processing, (ii) model inference and (iii) post-processing. In the pre-processing step, which is shared between methods, tissue

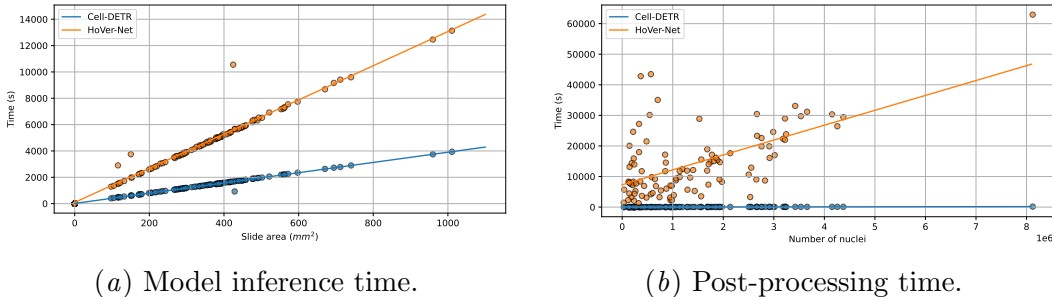

(*a*) Model inference time.   (*b*) Post-processing time.

Figure 2: Model inference and post-processing times as function of the slide area and the number of nuclei.

Cell-DETR is ×3.4 faster than HoVer-Net for inference, leading to significant differences when the area of the slide is large. Additionally, in contrast to HoVer-Net, the post-processing time of Cell-DETR is constant with respect to the number of detected nuclei.

regions are segmented from the thumbnail of the original WSIs and tiles of size $2048 \times 2048$ are extracted, ensuring that only tissue areas are processed by the model. An average of $1,300$ tiles are extracted for slide. For model inference, we employ the *window detection* approach for Cell-DETR whereas we follow the official processing pipeline for HoVer-Net. Figure 2(*a*) shows the model inference time for HoVer-Net and Cell-DETR as function of the total area of the slide. Notably, Cell-DETR shows a ×3.4 faster performance in this step, taking an average time of 1450s per slide, versus 4912s required by HoVer-Net. These metrics are extracted utilizing four 16GB GPUs. As previously argued, segmentation methods are more computationally demanding due to their dense output nature, making detection models a more suitable solution for inference on large WSIs. Finally, the post-processing step of Cell-DETR basically consists of combining the predictions of multiple tiles, while HoVer-Net requires an expensive post-processing to firstly obtain the instance segmentation masks from the raw predicted maps, and then to extract the cell nuclei instance information such as the centroids. Figure 2(*b*) shows the post-processing time as function of the number of nuclei detected in the slide. Intuitively, the post-processing time of HoVer-Net increases with the number of nuclei, which is in the order of millions for the WSIs. Instead, Cell-DETR exhibits a virtually constant post-processing time.

## 5. Conclusions

In this manuscript, we introduce a novel perspective to applications involving cell-level information on Whole Slide Images (WSIs), moving beyond conventional cell segmentation methods to prioritize detection while addressing reliability and scalability. Firstly, through a meticulous examination of design components, we enhance trustworthiness, achieving state-of-the-art performance in cell detection and classification that outperforms semantic segmentation methods. Secondly, we effectively tackle scalability challenges associated with large histopathological images, extending our approach to process WSIs with a remarkable efficiency. Consequently, our work provides vital insights for the development of diagnostics and interpretability applications, leveraging the wealth of information within extensive histopathology slides at the cellular level.

## Acknowledgments

This work has been supported by the Spanish Research Agency (AEI) under project PID2020-116907RB-I00 of the call MCIN/ AEI /10.13039/501100011033 and the FI-AGAUR grant funded by Direcció General de Recerca (DGR) of Departament de Recerca i Universitats (REU) of the Generalitat de Catalunya.

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

## Appendix A. Implementation details

Table 3 presents the image and window sizes used during the training and evaluation phases for the three datasets included in our experiments. For the PanNuKe dataset with $256 \times 256$-pixel images, no window partitioning is required for both training and evaluation. CoNSeP dataset images are larger at $1000 \times 1000$ pixels. Given the potential abundance of nuclei in a $1000 \times 1000$ image, we adopt the *window detection* procedure. Training involves random crops of size $250 \times 250$ from the original image, with one crop sampled per image at each epoch. During inference, we process the large images using sliding windows of size $250 \times 250$ and a stride of 187.

To merge predictions from overlapped sliding windows during inference, only detections whose centroid falls within the central crop of the window are considered. Specifically, for a window size of $250 \times 250$ and a stride of 187, the size of the selected central crop is $187 \times 187$, leaving a border of 61 pixels. Detections within these borders are excluded, as they will belong to the central crop of the neighboring window. For windows at the tile borders, this exclusion applies solely to the sides overlapped by another window, not to those corresponding to the image borders.

For the Camelyon16 dataset, lacking cell annotations for training, we utilize it to assess the scalability of cell detection and classification on Whole Slide Images (WSIs). The original images are divided into patches of size $2048 \times 2048$, processed similarly to CoNSep images, with a sliding window of size $256 \times 256$ and a stride of 187.

All our models are implemented in PyTorch, with hyperparameters drawn from the original Deformable DETR (Zhu et al., 2020), avoiding an exhaustive hyperparameter search. Training is performed on four NVIDIA Quadro RTX 16GB GPUs. The base learning rate, defined for a batch size of 16 in the original paper, is linearly scaled based on our setting. Multi-step learning rate scheduling is incorporated by a factor of 0.1 at 70% and 90% of training. Notably, for CoNSeP, the number of epochs is extended due to the limited number of training images and the use of only one crop of size $250 \times 250$ sampled from the entire image at each epoch, resulting in only 1/16 of the image being fed into the model.

Table 3: Window detection hyperparameters.

| Dataset | Training | | Evaluation and Inference | | |
|---|---|---|---|---|---|
| | Image size | Crop size | Patch size | Window size | Stride |
| **PanNuke** | 256 | 256 | 256 | 256 | - |
| **CoNSeP** | 1000 | 250 | 1000 | 250 | 187 |
| **Camelyon16** | - | - | 2048 | 256 | 187 |

## Appendix B. Evaluation metrics

The evaluation protocol for nuclei detection and classification follows the methodology outlined in (Graham et al., 2019), employing F1-score as the evaluation metric for enhanced

Table 4: Training hyperparameters.

| Dataset | Solver | | | | | Matcher | | | Loss | | | |
|---|---|---|---|---|---|---|---|---|---|---|---|---|
| | *epochs* | *base lr* | *batch size* | *lr drop* | *lr steps* | $\lambda_{giou}$ | $\lambda_{bbox}$ | $\lambda_{focal}$ | $\lambda_{giou}$ | $\lambda_{bbox}$ | $\lambda_{focal}$ | $\alpha_{focal}$ |
| **PanNuke** | 100 | 2e-4 | 2 | 0.1 | 70, 90 | 2 | 2 | 5 | 2 | 1 | 5 | 0.25 |
| **CoNSeP** | 1000 | 2e-4 | 2 | 0.1 | 700, 900 | 2 | 2 | 5 | 2 | 1 | 5 | 0.25 |

comparability. Initially, a bi-partite matching process aligns ground truth nuclei centroids with detected counterparts, limited to a radius of 12 pixels. Detection metrics, including true positives ($TP_{det}$), false positives ($FP_{det}$), and false negatives ($FN_{det}$), are derived based on the outcomes of the matching process between ground truth and predicted nuclei. The detection F1-score ($F_{det}$) is computed as the harmonic mean of detection precision ($P_{det}$) and recall ($R_{det}$).

For classification, $TP_{det}$ is further categorized into correctly and incorrectly classified nuclei of class $c$, denoted as $TP_c$ and $FP_c$, respectively. Additionally, misclassified elements from class $c$ are captured as $FN_c$. Precision, recall, and F1-Score for each class are then calculated as follows:

$$F_c = \frac{2(TP_c + TN_c)}{2(TP_c + TN_c) + 2FP_c + 2FN_c + FP_{det} + FN_{det}} \tag{1}$$

$$P_c = \frac{TP_c + TN_c}{TP_c + TN_c + 2FP_c + FP_{det}} \tag{2}$$

$$R_c = \frac{TP_c + TN_c}{TP_c + TN_c + 2FN_c + FN_{det}} \tag{3}$$

## Appendix C. Ablations

### C.1. Backbone feature levels and resolution

Deformable DETR enables multi-scale input features extracted from the backbone to boost the capabilities of the model. By default, Deformable DETR extracts three feature levels from the backbone and adds another virtual level on top of them with convolutional layer of kernel size 3 and a stride of 2. The first level is extracted from the second block of the backbone with a resolution of 1/8, and the remaining two levels are at 1/16 and 1/32.

The histopathology images employed in this work are scanned at a ×40 magnification, with a resolution of $0.245\mu m/px$. Given small and possibly enlarged shape of cell nuclei, the width or height of some instances can be of no more than 10px. Consequently, if the first feature level is extracted at 1/8, these small objects could be occluded in the backbone output representations. Table 5 shows the performance according on the first split of the PanNuke dataset for different configurations of the backbone and the output features. Generally, the Swin transformer backbone performs better than the ResNet50, accentuating the relevance transformer architectures for the medical image analysis. It can also be observed an increase on the F-Score, with larger margins in those nuclei types that are smaller, such as inflammatory cells and necrosis.

Table 5: Performance with distinct backbones.

| Backbone | Output levels | Detection | Neoplastic | Epithelial | Inflammatory | Connective | Necrosis |
|---|---|---|---|---|---|---|---|
| **ResNet50** | $1/8, 1/16, 1/32$ | 0.81 | 0.69 | 0.68 | 0.57 | 0.52 | 0.35 |
| **ResNet50** | $1/4, 1/8, 1/16, 1/32$ | 0.81 | 0.69 | 0.69 | 0.59 | 0.52 | 0.36 |
| **Swin** | $1/8, 1/16, 1/32$ | 0.82 | 0.72 | 0.72 | 0.59 | 0.55 | 0.42 |
| **Swin** | $1/4, 1/8, 1/16, 1/32$ | 0.82 | 0.73 | 0.74 | 0.61 | 0.55 | 0.45 |

## C.2. Loss function

Object detection loss for DETR involves predicting the bounding box centroid $(c_x, c_y)$ as well as the size $(w, h)$. Nonetheless, as mentioned in (Huang et al., 2023), for cell detection it is enough to predict the centroid of the cells. Indeed, the evaluation metrics only take into account the centroid, not the bounding box. In this section we explore the influence of including the boxes size information in the target. Results of Table 6 show a slight classification performance decline when the generalized intersection over union loss as well as the $(w, h)$ values of the L1 regression loss are excluded from the overall loss computation. Although it is information that may be ignored during inference, the supervision signals generated by their prediction could be providing valuable feedback to the network that may help in disambiguating the predictions for the multiple queries. Additionally, the generalized intersection over union is included as the L1 loss is highly influenced by the scale of the object (Carion et al., 2020). If removing this term, the model learns to focus on the bigger nuclei to minimize the loss function.

Table 6: Performance with and without bounding box size in the loss function.

| Target | Detection | Neoplastic | Epithelial | Inflammatory | Connective | Necrosis |
|---|---|---|---|---|---|---|
| $(c_x, c_y)$ | 0.82 | 0.71 | 0.73 | 0.59 | 0.55 | 0.42 |
| $(c_x, c_y, w, h)$ | 0.82 | 0.73 | 0.74 | 0.61 | 0.55 | 0.45 |

## Appendix D. Where is the model attending to?

The deformable attention mechanism only focuses on a subset of points for a given location, allowing for a detailed examination of the attended points within the predicted bounding boxes. In Figure 3 we present the detections and attention maps in both low and high cell density regions (Figure 3(a) and Figure 3(b), respectively). Concretely, we show the detected nuclei and bounding box (left), the attended sampling locations for each detection colored by the corresponding attention weight (middle) and finally the sampling locations colored by the head (right). It can be observed that the sampling locations are uniformly distributed along the bounding box. However, as depicted in the right images each head focuses on a specific direction within the bounding box. This phenomenon may be attributed to the ellipsoidal shape of cell nuclei and their frequent orientation.

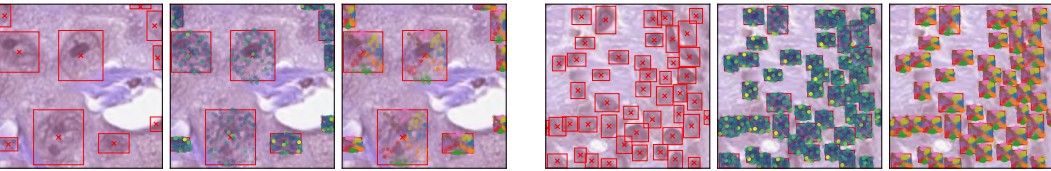

(*a*) Detections on a sparse region.      (*b*) Detections on a dense region.

Figure 3: Deformable attention maps.

Deformable attention maps show that different heads have learned to look at distinct directions.

