# OpenReview forum: "Cell-DETR: Efficient cell detection and classification in WSIs with transformers"
_MIDL.io/2024/Conference — MIDL 2024 Poster_

### Official Review · Reviewer_kD6i · 2024-02-18

**Confidence:** 4
**Preliminary Rating:** 2
**Final Rating:** 3.5

**Summary:**

The presented work investigates the application of the DETR model to the task of nuclei detection and classification with a focus on improving the inference time for WSIs. The authors evaluate their method on two well established benchmarks showing SOTA performance. Additionally, they demonstrate the scalability of their method on a WSI dataset and compare the results to a well-established method from the literature showing superior inference time with a 3.4$\times$ speed up factor.

**Strengths:**

- The authors address the clinically relevant task of detecting and classifying nuclei and demonstrate the SOTA performance by prioritizing a detection framework over more conventional segmentation methods, overcoming some of the disadvantages associated with these methods, such as higher computational cost during training and inference, and the need for more complex post-processing techniques.
- Furthermore, the development of a more efficient inference pipeline for application to WSIs provides a further step towards integration into routine clinical workflow.

**Weaknesses:**

- Although the authors claim to have conducted a thorough investigation of the design components of the DETR model, they have only used two different backbones, namely ResNet50 and SwinL, without justifying this choice or interpreting the results. There are no experiments with other components of the model to support the claim.
- The authors claimed that robustness is a focus of the method. However, there are no experiments to evaluate the robustness of the method such as a generalization to another dataset.
- While the comparison of the Cell-DETR with the window detection procedure showed a 3.4 $\times$ speed up in inference time compared to the Hover-Net inference pipeline is an interesting result, it does not show the speed up performance of the in-device sliding window approach itself, but rather that segmentation approaches are computationally more demanding than object detection approaches which was already mentioned by the authors. As such, the conclusions about the efficiency of the window detection procedure can not be taken seriously from this comparison. The efficiency of the window detection procedure should be compared using the same Cell-DETR model with a standard sliding window approach to make fair claims about the procedure.
- The authors report only single shot results which could be the result of an outlier or overfitting on the datasets. Repeating the experiments and reporting average results and standard deviations would provide a better understanding of the reliability of the results.

**Detailed Comments:**

- The authors claimed that they enhanced trustworthiness, however there are no experiments that would prove the reliability of the model beyond the in-domain results of the PanNuke and CoNSeP datasets.
- The interpretability of the model is demonstrated by the visualization of the deformable attention maps in chapter 4.3, but the interpretation of these maps is subjective and not an indication of model performance that would contribute to more trust in a model.

Minor comments:
- Typo in 4.2 “predictedmaps”
- Typo in 4.3 “that the the sampling”

**Justification Of Final Rating:**

The authors were able to clarify the issues regarding the reliability of the results on the investigated datasets. However, the investigation of the design components, in particular the experiments with different feature resolution levels and removed loss components is not very convincing in the context of small object detection.

**Justification Of The Preliminary Rating:**

As the efficiency of the method is a major claim of the paper, the comparisons should be fair to make meaningful conclusions. Furthermore, reporting only results of a single run can have major implications on reproducibility of the results.

**Questions To Address In The Rebuttal:**

- What is the speed up performance of the in-device sliding window approach compared to the standard sliding window approach using the same model?
- How reliable are the results when they are repeated with multiple random weight initializations?

---

> ### Author Response · Authors · 2024-03-14
> **Response to reviewer kD6i (I)**
>
> Dear reviewer,
>
> We appreciate your time and the opportunity to respond to your feedback and address the points raised in your review.
>
> 1. Speed-up performance
>
> We have analyzed a subset of the slides to measure the speed-up performance of the in-device sliding window approach in comparison to the standard sliding window approach, using the same model and configuration. Specifically, we observe a $\timesx1.1$ speed-up in inference time (Figure 2a) when slides are processed with the window detection pipeline compared to the traditional sliding window. Furthermore, we achieve a $\times2$ speed-up in post-processing time (Figure 2b). Notably, this acceleration is predominantly attributed to the reduction in data loading time facilitated by the window detection procedure. The approach minimizes the frequency of disk access and optimizes communication between the CPU and GPU.
> Our primary focus in the paper is to demonstrate the effectiveness and efficiency of cell detection transformers, advocating for cell detection over segmentation for tasks involving cell-level information in WSIs. Consequently, our time performance comparison is directed towards segmentation methods. Figure 2 serves as a valuable resource, providing substantial insights that justify our emphasis on cell detection and our chosen approach. We believe that the observed speed-up, especially in data loading time, reinforces the efficiency gains achieved by our in-device sliding window approach and substantiates our approach's relevance in the context of cell detection in WSIs.
>
> 2. Random weight initializations
>
> Thank you for bringing up this point. In our experiments, the weights of the DETR are not randomly initialized but rather pre-trained on COCO (Common Objects in Common) dataset, as traditionally done in the object detection literature. The only randomly initialized weights are the linear layers that project the hidden queries to the classification scores, as well as the neck connecting the backbone with the deformable transformer, which would suppose really slight variations in the final performance.
>
> Certainly, we have not properly mentioned initialization with pre-trained weights in our paper, which has also been pointed out by another reviewer. Consequently, we have specified the pre-training of the models in Section 3.2 Cell-DETR, concretely in the “Architecture” paragraph, by adding the following line to the revised manuscript in Section 3.2:
>
> >__Architecture.__ *[...]Both the backbone and the deformable transformer are pre-trained on the COCO dataset \cite{lin2014microsoft}[...]*
>
> Nonetheless, in order to address your concern regarding the reliability of the models with distinct initializations, we have run a set of experiments taking random weight initializations with different random generator seeds. As we’ve taken no pre-trained weights (not even for the backbone), the metrics are slightly lower, but we can observe that the standard deviation of the performance is minimal, validating the robustness and that the models actually converge to an optimal solution.
>
> - __detection__: 79.89 $\pm$ 0.32
> - __neoplastic__: 68.86 $\pm$ 0.19
> - __inflammatory__: 60.47 $\pm$ 0.46
> - __connective__: 53.29 $\pm$ 0.16
> - __necrosis__: 32.54 $\pm$ 2.29
> - __epithelial__: 66.7 $\pm$ 0.56

---

> > ### Author Response · Authors · 2024-03-14
> > **Response to reviewer kD6i (II)**
> >
> > 3. Design components
> >
> > Thank you for your observation regarding the choice of backbones and the perceived limitation in the investigation of design components. We appreciate the opportunity to provide further insight into our methodology.
> > The selection of Swin Transformer and ResNet50 as the two chosen backbones for our study is grounded in their widespread use within the DETR family for object detection, as evident in the related literature. Importantly, the decision is also influenced by the availability of publicly available weight initializations pre-trained on the COCO dataset, as highlighted in a previous comment. Swin-L transformer backbone increases the parameter complexity, but, in contrast, offers a superior performance with respect to the traditional ResNet-50. These results align with the widespread adoption of transformers as general purpose architecture within the deep learning community.
> >
> > Beyond the choice of backbones, our investigation extends to various components of the DETR model. We explored the influence of the number of feature levels output by the backbone and the resolution of these feature maps. Our findings highlight the necessity of higher resolutions for accurate detection and classification of smaller cell nuclei, such as inflammatory cells.
> >
> > Additionally, we delve into experiments on elements of the loss function. While some related work simplifies cell nuclei representation to a centroid, our experiments indicate a decrease in performance when removing the bounding box size from the representation. This emphasizes the significance of including box size information in disambiguating detections and providing additional feedback for classification.
> >
> > Although these observations are outlined in Section 3.2 of the paper, we acknowledge the constraints of length and have reported the detailed results in the Appendix C. We firmly believe that these experiments, coupled with the extensive exploration of different scenarios (small patches, tiles and WSIs), provide a thorough understanding of DETR's applicability for cell nuclei detection and classification. We appreciate your feedback and hope this clarification addresses your concerns.
> >
> >
> > 4. Robustness
> >
> > Thank you for bringing up the issue of robustness. While we recognize the significance of evaluating generalization to other datasets as a metric of robustness, we'd like to emphasize that the PanNuke dataset itself, being one of the most common benchmarks in the literature, poses considerable challenges for nuclei detection and classification.
> >
> > The PanNuke dataset, known for its challenging nature with samples from 19 different tissues and imbalances between tissues and cell nuclei classes, is indeed a rigorous benchmark. In our study, we adopted the original paper's 3-fold split for the dataset, ensuring a thorough evaluation by reporting average metrics across folds after training and validating with the other two. The results presented in Table 1 of our paper showcase the model's effectiveness across this challenging dataset, illustrating its robustness in handling diverse tissue types and imbalanced class distributions.
> >
> > While we understand the importance of cross-dataset generalization, we believe that demonstrating robustness on the PanNuke dataset, as a representative challenging dataset for nuclei detection and classification, as well as our results on the CoNSep dataset, are a valuable contribution to the field that demonstrates the robustness of the method.
> >
> > We extend our gratitude for your valuable feedback and look forward to any further discussions or clarifications that may arise.

---

> > > ### Comment · Reviewer_kD6i · 2024-03-21
> > > **Response to the authors**
> > >
> > > Dear authors,
> > > Thank you for your thoughtful response to my review.
> > >
> > > 1.	Speed-up performance
> > >
> > > With regards to the actual speed-up gains of the in-device window detection approach the authors analyzed a subset of slides and found their in-device sliding window approach to be 1.1x faster than the standard approach for inference and 2x faster for post-processing. This speed-up is mainly due to reduced data loading time achieved by their method. They acknowledge their focus is on cell detection over segmentation and believe Figure 2 justifies this focus due to the efficiency gains observed. I appreciate the effort taken by the authors to clarify my concerns and acknowledge the authors focus on a comparison with segmentation approaches.
> > >
> > > 2.	Random weight initializations
> > >
> > > The authors acknowledge not mentioning pre-training in the paper and have added a line specifying pre-training on COCO. Additionally, they ran new experiments with random weight initialization and found minimal performance variation. While the minimal variation with random seeds is valuable, my original question was poorly phrased. I was hoping to see multiple repetitions of the experiments that led to the results in the paper itself.  This would provide a better understanding of the overall reproducibility of those results. While we now have an overview of the variation with random initialization, we still don’t know whether the results reported in the paper are outliers or reliable.
> > >
> > > 3.	Design components
> > >
> > > The authors justify their backbone choices (Swin Transformer and ResNet50) by mentioning their prevalence in DETR and pre-trained weight availability. They also acknowledge exploring other design components (feature levels, resolution, loss function) in the appendix to validate their overall approach. While the experiments provide valuable context, a  deeper exploration within the appendix might strengthen the case for a thorough investigation. For instance, the appendix could delve deeper into the impact of specific removed loss function components on performance, going beyond confirming the importance of bounding box information. Likewise, exploring the impact of parameter allocation across feature levels (e.g., increasing parameters in higher resolutions) would provide more novel insights than confirming the benefit of higher resolutions for smaller objects.
> > >
> > > 4.	Robustness
> > >
> > > The authors acknowledge the importance of evaluating on other datasets but argue that PanNuke's inherent challenges (diverse tissues, imbalanced classes) make it a strong benchmark for robustness. While open to cross-dataset evaluation in future work, they believe PanNuke and CoNSep results already showcase robustness. While I acknowledge the importance of public benchmark datasets claiming robustness should refer to performance evaluation under certain distribution shifts that can arise during inference on unseen image domains. The PanNuke dataset is splitted randomly into train, val and test set and hence distributions are similar and a model trained on the train might perform well on the test split but the performance on unseen images (e.g. from another center, another organ) is completely unknown.

---

> > > > ### Author Response · Authors · 2024-03-22
> > > > **Response to comment (I)**
> > > >
> > > > Dear reviewer,
> > > > Thank you for your insightful comments and feedback on our review. We appreciate the opportunity to address your concerns and provide clarification on various aspects of our methodology. Below, we have responded to each of your points in detail, aiming to provide a comprehensive understanding of our approach and its evaluation. We hope that our responses effectively address your queries and contribute to a deeper appreciation of our work. Should you require further information or clarification, please do not hesitate to let us know. We value your engagement and interest in our research.
> > > >
> > > > 2. Random weight initializations
> > > >
> > > > We appreciate your clarification regarding the expectation for multiple repetitions of experiments within the paper itself to gauge the reproducibility of our results. We understand the importance of ensuring the reliability of reported findings by running multiple iterations.
> > > >
> > > > To address your concern, it's crucial to emphasize that our experiments are not conducted with random weight initializations; instead, we utilize pre-trained weights, ensuring consistency across experiments. This pre-training on the COCO dataset forms the basis for our experiments and guarantees a standardized starting point for each run.
> > > > To assess the reliability and robustness of our results, we incorporate multiple iterations with different data partitions. As detailed in Section 4.1 of the paper, the reported metrics in Table 1 represent averages across three standard splits of the PanNuke dataset. Each iteration follows a distinct training-validation-testing setup, providing a comprehensive evaluation framework:
> > > >
> > > > >Table 1 presents the detection and classification metrics on the PanNuke dataset for Cell-DETR using ResNet-50 and Swin Transformer (large) as backbones, in comparison to other state-of-the-art segmentation and detection methods. The provided numerical values in the table __detail the averaged precision (P), recall (R), and F1-score (F1) for detection and nuclei types across the three standard splits publicly available for this dataset__.
> > > >
> > > > We apologize for any confusion regarding this matter and appreciate your feedback, which has prompted us to clarify this aspect in the final version of the paper. If you have any further questions or concerns, please feel free to let us know.
> > > >
> > > > 3. Design components
> > > >
> > > > Thank you for your understanding and acknowledgment of the importance of exploring the influence of loss function components in understanding our method better. Indeed, our exploration of the bounding box size's influence effectively assesses this aspect, particularly in relation to the removal of the intersection over union term. he loss function of DETR-like models has three main components: bounding box regression, object classification and intersection over union. Bounding box regression and object classification cannot be removed, as it would be removing the supervised loss (regression and classification). As for intersection over union, it is included to overcome the scale-dependent L1 box regression problem. When we remove the bounding box size (w,h) from the target, note that we modify the L1 loss function to only include (cx, cy), and the intersection over union term is completely removed.
> > > >
> > > > Regarding the architecture-based components, we have extensively examined critical design elements such as backbone selection and the number of feature levels and resolutions. These components are pivotal for any object detection task and particularly crucial for cell detection, given the inherent challenges posed by potential overlap and the small size of cell nuclei.
> > > >
> > > > However, we would appreciate further clarification on your query regarding "parameter allocation across feature levels (e.g., increasing parameters in higher resolutions)." If you are referring to the dimensionality of tokens at each resolution, it's essential to note that the Swin Transformer backbone comes with a fixed size specific to every resolution or level. Modifying these values would impede the use of the pre-trained backbone. If there are any other aspects you would like us to address or if further clarification is needed, please feel free to let us know.

---

> > > > > ### Author Response · Authors · 2024-03-22
> > > > > **Response to comment (II)**
> > > > >
> > > > > 4. Robustness
> > > > >
> > > > > We acknowledge the importance of cross-dataset validation and appreciate your interest in this aspect. We understand your concerns regarding the PanNuke dataset's random split.
> > > > > In response to your feedback, we have downloaded, processed and evaluated on the MoNuSeg dataset (https://monuseg.grand-challenge.org/). This dataset does not include cell classification annotations, but only segmentation masks (that are converted to bounding boxes), consequently, only the detection performance can be addressed. The images are sized 1000x1000 pixels, so that the window detection procedure has been employed for the evaluation. With no fine-tuning on the dataset, we have obtained a 81.25% of F1-Score on cell nuclei detection. We believe that these results showcase the robustness of our method.
> > > > >
> > > > > We trust that these responses address your concerns, and we remain available to provide further information or clarification as needed. Thank you for your continued interest in our work.

---

### Official Review · Reviewer_jWp8 · 2024-02-27

**Confidence:** 4
**Preliminary Rating:** 4
**Final Rating:** 4

**Summary:**

The authors use detection transformers for cell detection and classification, on Whole Slide Images. The models show a significantly faster inference time on large WSIs, which is suitable for applying to real world scenarios.

**Strengths:**

- The authors have undertaken extensive experiments to find the perfect components of DETR models.
- A special pipeline is designed for accelerating the procedure of detection on cells in WSIs.
- Good empirical performance

**Weaknesses:**

- Could the authors explain more of the characteristics of WSIs, and the challenges that it poses to the existing detection models?
- Since the authors are using Transformers as the backbone of detection models, could the authors explain what sorts of pretrained weights are used and why?
- The authors mentioned the technique of cropping images into overlapping areas, in case of large images. However, if the objects placed in overlapped areas are bounded differently in neighboring results, how to render the final result?
- Could the authors show the number of trainable parameters and FLOPS of the method?
- The authors mentioned a ‘comprehensive exploration’ on detection transformers for cell detection and classification. However it is only applying sliding cropping in large images. It seems to lack novelty.

**Detailed Comments:**

- Could the authors explain more of the characteristics of WSIs, and the challenges that it poses to the existing detection models?
- Since the authors are using Transformers as the backbone of detection models, could the authors explain what sorts of pretrained weights are used and why?
- The authors mentioned the technique of cropping images into overlapping areas, in case of large images. However, if the objects placed in overlapped areas are bounded differently in neighboring results, how to render the final result?
- Could the authors show the number of trainable parameters and FLOPS of the method?
- The authors mentioned a ‘comprehensive exploration’ on detection transformers for cell detection and classification. However it is only applying sliding cropping in large images. It seems to lack novelty.

**Justification Of Final Rating:**

I would like to extend my sincere thanks to the authors for their comprehensive classification work. The incorporation of further experiments and the broadening of classifications have thoroughly addressed all of my concerns. Furthermore, I have discovered a piece of similar work titled "Class-aware adversarial transformers for medical image segmentation," which employs the transformer architecture and a sampling rule. I suggest that the authors consider discussing this in their final revision.

====update=====  Overall, I would like to vote for acceptance.

**Justification Of The Preliminary Rating:**

- Could the authors explain more of the characteristics of WSIs, and the challenges that it poses to the existing detection models?
- Since the authors are using Transformers as the backbone of detection models, could the authors explain what sorts of pretrained weights are used and why?
- The authors mentioned the technique of cropping images into overlapping areas, in case of large images. However, if the objects placed in overlapped areas are bounded differently in neighboring results, how to render the final result?
- Could the authors show the number of trainable parameters and FLOPS of the method?
- The authors mentioned a ‘comprehensive exploration’ on detection transformers for cell detection and classification. However it is only applying sliding cropping in large images. It seems to lack novelty.

**Questions To Address In The Rebuttal:**

- Could the authors explain more of the characteristics of WSIs, and the challenges that it poses to the existing detection models?
- Since the authors are using Transformers as the backbone of detection models, could the authors explain what sorts of pretrained weights are used and why?
- The authors mentioned the technique of cropping images into overlapping areas, in case of large images. However, if the objects placed in overlapped areas are bounded differently in neighboring results, how to render the final result?
- Could the authors show the number of trainable parameters and FLOPS of the method?
- The authors mentioned a ‘comprehensive exploration’ on detection transformers for cell detection and classification. However it is only applying sliding cropping in large images. It seems to lack novelty.

---

> ### Author Response · Authors · 2024-03-14
> **Response to reviewer jWp8 (I)**
>
> Dear reviewer,
>
> We appreciate your valuable feedback and are pleased to provide a comprehensive response addressing the points raised in your review.
>
> 1. Characteristics of the WSIs
>
> WSIs are high-resolution, digital representations of an entire histopathological slide containing a tissue sample. These samples are stained and then scanned at a resolution that captures microscopic details. The scale of these slides is giga-pixel (ie. 100,000 x 100,000 = 10^10 pixels), so that the implementation of any computer-aided diagnosis becomes non-trivial. This challenge is augmented for segmentation methods, as the outputs are dense pixel-level maps, leading to giga-pixel output maps.
>
> For a better understanding of this idea in the text, we’ve included the following in the introduction:
>
> >*Despite the valuable boost in accuracy, [...]. The size of WSIs, often reaching gigapixel dimensions (e.g. $100,000\times100,000=10^10$ pixels), adds an extra layer of complexity to implementing a computer-aided diagnosis pipeline, requiring the partition into smaller patches and subsequent processing. This challenge is particularly pronounced for segmentation methods, given their dense pixel-level output maps, making them impractical for real-world applications on WSIs. Instead, their application is often limited to smaller tiles, hindering the development of computer-aided diagnoses that require comprehensive cell information.*
>
> Additionally, in the description of the datasets, we have included the actual average size of the slides processed, as well as the percentage of tissue area and the average number of tiles extracted per WSI, so that the reader has an exact estimation of the actual computational challenge:
>
> >__Camelyon16.__ *The Camelyon16 dataset (Bejnordi et al., 2017) consists of 400 H\&E stained Whole Slide Images (WSIs) of lymph node sections scanned at $\times 40$ magnification. Each WSI is accompanied by annotations highlighting tumor and normal regions. With average dimensions of $189,832 \times 95,590$px, approximately $29 \%$ of the slides represent tissue area. Specifically, an average of 1384 tissue tiles, each sized $2048\times2048$ pixels, is extracted per slide. Despite lacking cell-level annotations for quantifying detection and classification performance, this dataset remains pivotal in evaluating the efficacy of our models.*
>
> 2. Pre-trained weights
>
> Thank you for bringing up this point. Indeed, it is not properly mentioned in the manuscript. In our work, both the Deformable DETR and the backbones are initialized with the pre-trained weights on the COCO (Common Objects in Common) dataset. This practice is common in object detection, as new approaches are typically benchmarked on COCO, and authors often release their pre-trained weights for wider use. We have specified the pre-training of the models in Section 3.2 Cell-DETR, concretely in the “Architecture” paragraph, by adding the following line:
>
> > __Architecture.__ *[...]Both the backbone and the deformable transformer are pre-trained on the COCO dataset (Lin et al., 2014).[...], highlighted in blue in the revised manuscript.*
>
> 3. Combining predictions
>
> In our implementation, we acknowledge the possibility of overlapping areas when cropping large images and the potential variation in bounding results. To address this, we leverage the compact nature of cell nuclei and employ a specific strategy during the window detection procedure.
> For each window of size 256x256 px with a stride of 192 px (resulting in a 64 px overlap between neighboring windows), we consider only those detections whose centroid falls within the central crop of each window. Specifically, for a window size of 256x256 px, the central crop is defined as 192x192 px. Consequently, detections within the outer margin of 32 px (64/2) of the window are masked out. This ensures that only detections within the central region, minimizing potential inconsistencies at the borders of overlapping areas, are considered for the final result.
> It's important to note that for windows located at the borders of the original, larger image, the corresponding margin is not masked. This approach helps maintain the integrity of detections at the image borders while effectively handling overlapping areas during the cell nuclei detection process.
>
> Although it is mentioned in section 4.1 for the CoNSeP dataset, we have included a more detailed explanation of this step in the Implementation details section in the appendix in the revised manuscript. We appreciate your attention to this detail, we hope that this explanation contributes to a better understanding of the method and its effective handling of overlapping areas during the cell nuclei detection process.

---

> > ### Author Response · Authors · 2024-03-14
> > **Response to reviewer jWp8 (II)**
> >
> > 4. Parameters and FLOPs
> >
> > In response to your request for information on trainable parameters and FLOPs of our method, we've provided a breakdown for both backbone models (Swin-L and ResNet50) and the remaining components. It's noteworthy that the implementation of Deformable Attention offers a scalable solution that is more optimal than standard self-attention. Although the Swin-L backbone contributes significantly to the number of parameters and FLOPs, this backbone delivers superior performance, still being much more optimal than segmentation methods and providing a reliable, scalable solution.
> >
> > | Component              | Params (M) | FLOPs (G) |
> > |------------------------|------------|-----------|
> > | Backbone (Swin-L)     | 194        | 60        |
> > | Backbone (R50)        | 23         | 5         |
> > | Transformer + output  | 12         | 40        |
> >
> > The reported numbers are estimated using the Python [thop](https://github.com/Lyken17/pytorch-OpCounter) tool.
> >
> > 4. Comprehensive exploration
> >
> > Thank you for raising this concern. By including the term ‘comprehensive exploration’ in the abstract, we aim to emphasize the multifaceted nature of our work. In this work, not only have we adopted the window detection approach to enable the inference on larger images to overcome the limitations of detection transformers for digital pathology image analysis and evaluated the behavior of these models in different scenarios (small patches, larger image tiles and WSIs), but we have also conducted an exploration of the distinct design components of the DETR models such as the backbone, the number of feature levels in the pyramid, as well as the components of the loss function in order to obtain an accurate cell detection and classification. Given the length restriction of 8 pages, the tables showing the results are given in the appendix.
> >
> > Thank you for the insightful reviews. We trust that our response has successfully addressed your concerns and provided the clarity you needed.

---

### Official Review · Reviewer_VZVH · 2024-02-28

**Confidence:** 3
**Preliminary Rating:** 4

**Summary:**

The authors conduct an investigation into the suitability of detection transformers for cell detection and classification, referred to as Cell-DETR. Their study proves the method's efficacy by surpassing existing benchmarks but also devises a workflow capable of handling these tasks efficiently on Whole Slide Images at a large scale, for the development of subsequent applications.

**Strengths:**

Firstly, the authors have laid out a well-organized framework, beginning with a detailed explanation of the methodology employed, namely the detection transformers for cell detection and classification (Cell-DETR). This sets a solid foundation for understanding the subsequent analyses and results.

Secondly, the manuscript is structured in a manner that systematically addresses various aspects of the method's performance on large WSIs. It likely includes discussions on the computational complexities involved, such as memory usage and processing time, when dealing with vast image datasets.

**Weaknesses:**

Images and tables could be improved.
There is a lack on being specific. Examples: what means: "a significantly faster", in which way the work "pave the way for improved diagnostics", "practical solution", define:"high -density regions", "large images" (measures in pixels are presented on the results but a higher specification is still necessary). Being more specific would surely help to transfer the novelty of the results

**Detailed Comments:**

I think the manuscript is well presented and addresses an important issue, but being more specific would help the people to reuse and recite this work

**Justification Of The Preliminary Rating:**

Overall, the manuscript's emphasis on demonstrating the method's effectiveness and feasibility in the context of large WSIs suggests a thorough examination of the challenges inherent in such scenarios.
The manuscript, while well-presented and structured, may lack specific details that could potentially elevate its impact. Such specifications are crucial for providing a comprehensive understanding of the research and its implications.

**Questions To Address In The Rebuttal:**

Be more specific by using clear statements
Improve the images

---

> ### Author Response · Authors · 2024-03-14
> **Improving text and figures**
>
> Dear reviewer,
>
> Thank you for your valuable feedback. In response to your suggestion to enhance specificity, we have made the following changes to improve the readability and understanding of the manuscript. Although the changes are coloured in blue in the submitted revised version, we’re including them in this rebuttal:
>
> In the abstract, we now explicitly emphasize the efficiency of our proposed method, highlighting a x3.4 improvement in performance. Furthermore, to clarify the term 'improved diagnostics,' we have updated the abstract to convey our intent of developing diagnosis pipelines leveraging cell-level information.
>
> >*[...] We show its efficiency and feasibility by reporting a x3.4 faster inference time on a dataset featuring large WSIs. [...]. By addressing the challenges associated with large-scale cell detection, our work contributes valuable insights that paves the way for the development of scalable diagnosis pipelines based on cell-level information.*
>
> As for the challenges associated with larger images, we've revised __Section 3.3__ to elaborate on the constraints of extending DETR models to image tiles, such as those sized 1024x1024 or 2048x2048 pixels. By specifying the number of nuclei in small patches corresponding to high-density regions, we aim to provide a clearer understanding of the computational challenges involved.
>
> >*A significant constraint of DETR-like models is the necessity for the number of queries in the decoder to surpass the potential objects present in an image. In regions characterized by a high cell density, a $256\times256$px image patch may contain up to 300 cell nuclei. Consequently, increasing the input image size to larger tiles, such as $1024\times1024$px or $2048\times2048$px, becomes non-trivial. The number of cell nuclei, and therefore the required input DETR queries, can substantially increase, potentially resulting in prohibitive computational demands.*
>
> >*[...]. This strategy allows Cell-DETR to overcome the limitations associated with regions containing high density of cell nuclei and large images, ensuring the model's adaptability and efficiency in real-world applications.*
>
> Aligning with another reviewer's feedback, we've enhanced the description of WSIs throughout the manuscript. In the introduction, we now explicitly introduce the challenges of dealing with WSIs, setting the context for our work:
>
> > *Despite the valuable boost in accuracy, this improvement comes at a cost—significant computational demands during both training and inference. [...]. The size of WSIs, often reaching gigapixel dimensions (e.g. $100,000\times100,000=10^{10}$ pixels), adds an extra layer of complexity to implementing a computer-aided diagnosis pipeline, requiring the partition into smaller patches and subsequent processing. This challenge is particularly pronounced for segmentation methods, given their dense pixel-level output maps, making them impractical for real-world applications on WSIs. Instead, their application is often limited to smaller tiles, hindering the development of computer-aided diagnoses that require comprehensive cell information.*
>
> Furthermore, in the Camelyon16 dataset description, we've included details such as the average size and tissue area of the slides, along with the number of extracted patches. This addition serves to highlight the inherent challenges associated with working on WSIs
>
> > __Camelyon16.__ *The Camelyon16 dataset  (Bejnordi et al., 2017) consists of 400 H\&E stained Whole Slide Images (WSIs) of lymph node sections scanned at $\times 40$ magnification. Each WSI is accompanied by annotations highlighting tumor and normal regions. With average dimensions of $189,832 \times 95,590$px, approximately $29 \%$ of the slides represent tissue area. Specifically, an average of 1384 tissue tiles, each sized $2048\times2048$ pixels, is extracted per slide. Despite lacking cell-level annotations for quantifying detection and classification performance, this dataset remains pivotal in evaluating the efficacy of our models. Leveraging the Camelyon16 dataset allows us to assess the practicality of Cell-DETR on a scale that closely mimics the challenges encountered in clinical settings, demonstrating their scalability and effectiveness in handling large-scale pathology images.*
>
> We also appreciate your feedback concerning the figures, understanding their crucial role in aiding reader comprehension. In response, we have made enhancements to better emphasize the application of cell detection in WSIs with transformers. Specifically, Figure 1 has been updated to encompass the entire cell detection pipeline on WSIs, integrating preprocessing, inference, and post-processing stages. To maintain the paper's focus and adhere to page limits, we have relocated Section 4.3, "Where is the model attending to?", along with its corresponding figure, to the appendix. We believe these modifications contribute to a clearer understanding of our work.

---

> > ### Comment · Reviewer_VZVH · 2024-03-27
> > **Review's response**
> >
> > Dear authors,
> > I appreciate your throughtful response to my review. Some details are now clarified.
> > However, in order to reuse and recite this work the robustness need to be clarified. Please see the comment made by other reviewer, since I share the issue: "Stating robustness should refer to evaluating performance under certain distribution shifts that may arise during inference on unseen image domains."

---

> > > ### Author Response · Authors · 2024-03-27
> > > **Response to comment**
> > >
> > > Dear reviewer,
> > >
> > > Thank you for your feedback. We acknowledge the importance of cross-domain generalization when assessing robustness.
> > > To address this aspect, we conducted evaluations on the MoNuSeg dataset (https://monuseg.grand-challenge.org/), as commented to the other's reviewer.  This dataset lacks cell label and only includes segmentation masks, which are converted to bounding boxes. Consequently, we have addressed the detection performance. The images are sized 1000x1000 pixels, so that the window detection procedure has been employed for the evaluation. With no fine-tuning on the dataset, we have obtained a 81.25% of F1-Score on cell nuclei detection. We believe that these results showcase the robustness of our method under distribution shifts.
> > >
> > > We will ensure that these additional results are incorporated into the final version of the manuscript. We hope that these clarifications address your concerns and contribute to the overall quality of our work. Please feel free to reach out if you have any further questions or require additional information.
> > >
> > > Thank you once again for your valuable feedback.

---

### Meta-Review · Area_Chair_XreS · 2024-04-04

**Recommendation:** Accept (Poster)
**Confidence:** 4

**Metareview:**

The reviewers concur that the study holds relevance within the MIDL community. The authors' responses exhibit a convincing effort in addressing the reviewers' concerns comprehensively, providing ample justification for their methodological and experimental decisions. Their willingness to accommodate necessary clarifications or adjustments to fortify their work is evident. However, a noteworthy observation from one reviewer highlights the existence of a similar study titled "Class-aware adversarial transformers for medical image segmentation," which also employs transformer architecture and a sampling rule. Integrating discussion on this related work could enhance the paper's depth and offer a broader context for the research presented. Therefore, I urge the authors to consider addressing this aspect before the final submission. Considering these factors, I recommend accepting this work for publication in MIDL.

---

### Decision · Program_Chairs · 2024-04-06

Accept (Poster)